# Reduced levels of mitochondrial ribosomal protein *MRPL54* does not alter Apc related adenoma formation

**Claudia N. Spaan**[1,2☯], **Eileen Daniels**[3,4☯], **Wouter L. Smit**[1,2], **Ruben J. de Boer**[1,2], **Joana Silva**[5], **Jacqueline L. M. Vermeulen**[1], **S. Meisner**[1], **Vanesa Muncan**[1], **Riekelt H. Houtkooper**[3,4,6,7], **Jarom Heijmans**[1,2,4]*

**1** Amsterdam UMC, University of Amsterdam, department of Gastroenterology and Hepatology, Tytgat Institute for Liver and Intestinal Research, Amsterdam, Netherlands, **2** Cancer Center Amsterdam, Cancer Biology and Immunology, Amsterdam, The Netherlands, **3** Laboratory Genetic Metabolic Diseases, Amsterdam UMC location University of Amsterdam, The Netherlands, **4** Amsterdam Gastroenterology Endocrinology and Metabolism institute, Amsterdam, The Netherlands, **5** Department of Oncogenomics, Netherlands Cancer Institute, Amsterdam, The Netherlands, **6** Amsterdam Cardiovascular Sciences institute, Amsterdam, The Netherlands, **7** Emma Center for Personalized Medicine, Amsterdam UMC, Amsterdam, The Netherlands

☯ These authors contributed equally to this work.
* j.heijmans@amsterdamUMC.nl

## Abstract

Reprogramming of energy metabolism is one of the hallmarks of cancer cells and mutations that modify wild type intestinal cells to colon carcinomas increases cellular energy expenditure. Mitochondria are the main site for ATP production in (cancer) cells and disrupting their function results in impaired tumor forming efficacy. The mitochondrial ribosomal proteins (MRPs) constitute the ribosome specifically in mitochondria, and as such are crucial for the translation process of the electron transport chain complex subunits. We hence aimed to explore the consequence of reduced MRP expression on adenomagensis and investigate this in a genetic mouse model with bodywide heterozygosity for *Mrpl54*. We show that *Mrpl54* heterozygosity does not alter adenoma formation, intestinal proliferation or apoptosis in a heterozygous *Apc* model. Furthermore, diminished *Mrpl54* expression did not decrease stemness or global parameters of metabolism in colorectal cancer cell lines.

## Introduction

The observation by Otto Warburg that cancer cells have reprogrammed cellular metabolism dates back one century but more recently, in the seminal work of Hanahan and Weinberg on hallmarks of cancer, reprogramming of energy metabolism in cancer has been pointed out as an emergent hallmark [1,2]. In the development of colorectal cancer, mutations that result in the sequential development of adenomas and eventually invasive adenocarcinomas influence cancer metabolism [3–7]. The fact that increased protein translation and energy expenditure occurs in cancer cells,

**Data availability statement:** All relevant data are within the paper and its Supporting Information files.

**Funding:** This work was supported by a grant of the Dutch cancer foundation (KWF/Alpe 11053/2017-1) and by a grant of the Netherlands organisation for Scientific Research (NWO-Veni 91615032). The funders had no role in study design, data collection and analysis, decision to publish, or preparation of the manuscript.

**Competing interests:** The authors have declared that no competing interests exist.

but not in their non cancerous counterparts, indicates the presence of excess capacity in biological processes that govern translation and energy requirements which is only addressed during pathyphysiological circumstances. Moreover, reduced capacity for protein production and a reduced threshold for activation of the Unfolded Protein Response (UPR) due to intestinal epithelial heterozygosity for endoplasmic reticulum (ER) resident chaperone GRP78 exposed remarkable disparity between protection against tumorigenesis and absence of phenotypical alterations under homeostatic conditions [7,8]. These results corroborated presence of excess cellular capacity that is only addressed in specific pathological situations such as adenoma formation. This notion can lead to identification of molecular targets that may be inhibited to reduce tumorigenesis with little effects on non-cancerous cells could result in novel anticancer treatment strategies.

A crucial energy source is the generation of ATP by oxidative phosphorylation (OXPHOS) in the mitochondria, which consists of four enzyme complexes forming the respiratory chain and complex V as the terminal ATP synthase. Most of the OXPHOS complexes are partly encoded by nuclear DNA (nDNA) and partly encoded by mitochondrial DNA (mtDNA) and coordinated synthesis and assembly is required for optimal function. The mtDNA has a dedicated, distinct, transcription and translation machinery that is independent from equivalent processes in the nucleus and cytosol. mtDNA is translated by the mitochondrial ribosome, a protein complex that contains mitoribosomal proteins (MRPs). Thus, mtDNA and mitoribosomal proteins (MRPs) are critically important for biogenesis of the OXPHOS complexes. [9,10]. Therefore, a number of studies have focused on altering mtDNA or MRPs in cancer models [11]. Indeed, after depleting mtDNA, tumorigenic potential of cancer cells was diminished *in vitro* en *in vivo* [12–14]. In diverse cancers, such as melanoma, breast-, colorectal-, liver– and lung cancer, MRPs are upregulated and reduction of specific MRPs suppresses tumor growth [15–17].

MRPL54 is a highly conserved mitochondrial ribosomal protein that is ubiquitously expressed and is critical for mitochondrial translation and OXPHOS activity [18–20]. Previously, homozygous *Mrpl54* knockout mice proved to be lethal. Under homeostatic conditions, *Mrpl54* heterozygous knockout mice exhibited reduced *Mrpl54* gene expression, but showed no difference in body composition, lifespan and activity compared to wildtype controls [21,22]. Though *Mrpl54* heterozygosity has little influence on gross phenotypical characteristics, this may result from an abundant OXPHOS reserve capacity during homeostasis. The increased cellular growth and higher energy requirements during adenomaformation challenge the OXPHOS complex more compared to standard laboratory conditions. We propose therefore that genetically interfering with the OXPHOS complex can impact cells in a highly proliferative state. We set out to test the hypothesis that *Mrpl54* heterozygosity would hamper adenomaformation.

## Materials and methods

### Animal experiments

This study was approved by the nationally institutional animal welfare board (Central Committee Animal Experiments) of the Netherlands. Mouse experiments were

performed in the Academic Medical Center Animal Research Institute in accordance with local guidelines under protocol number ALC284. VillinCre[ERT2,] Rosa26[LacZ], Apc[fl] alleles were all described previously [23–27]. These mice were crossed to *Mrpl54* heterozygous mice [21]. All mice had a C57BL/6 background and were between 6–8 weeks old. All experimental groups contained an equal number of male and female mice. Animals were sacrificed using the inhalant anesthetics carbon dioxide ($CO_2$), followed by cervical dislocation.

## Adenoma studies

For Cre[ERT2] mediated recombination, mice were injected daily for 5 consecutive days with with 50 mg/kg tamoxifen (T5648, Sigma-Aldrich, St Louis, MO 10 mg/ml in corn oil). Two hours prior to sacrifice all mice received 100 mg/kg BrdU intraperitoneally (Sigma-Aldrich, St Louis, MO 10 mg/mL in NaCl.). Mice were sacrificed 20 weeks after Cre–mediated recombination. After sacrifice, intestines were immediately taken out and rinsed in cold PBS and the small intestine was divided in a proximal-, middle- and distal part.

Adenomas were counted macroscopically in the small intestine and colon. Investigators were blinded for mouse genotype during adenoma quantification and all adenoma quantification was performed by two investigators independently. After paraffin embedding (see section below), haematoxilin and eosin sections were generated from all parts of the murine intestine (proxima, middle, distal, colon) and quantification on microscopic sections was performed by two independent investigator, blinded for the murine genotype.

## Tissue preparation, immunohistochemistry and X-Gal staining

Tissue was fixed overnight in 4% buffered formaldehyde in PBS. Formaldehyde was replaced with 70% ethanol and processed according to standard protocols for paraffin embedding [28]. After paraffin embedding, 4,5 μm sections were generated using a microtome. Immunohistochemistry was performed as previously described [29]. In short, 4,5 μm sections were deparaffinized and rehydrated. Endogenous peroxidase was blocked in 0.3% $H_2O_2$ in methanol. For antigen retrieval, slides were treated at 96°C for 10 minutes in 10 mM sodium citrate buffer pH 6.0, or for 20 minutes in 10 mM Tris 1mM EDTA buffer pH 9.0 and incubated overnight at 4°C with primary antibody diluted in PBT (PBS with 0.1% Triton X-100 and 1% w/v BSA). Primary antibodies: anti-BrdU mouse monoclonal (Roche BMC9318) 1:500; anti-cleaved-caspase-3 (Cell Signaling 9661S) 1:400. Antibody binding was visualized with Powervision (Immunologic) and substrate development was performed using diaminobenzidine (Sigma-Aldrich D5637-10G, St Louis, MO). Hematoxylin was used as counterstain.

For assessment of Cre-mediated recombination, cells expressing the LacZ allele were visualized using X-Gal staining. To this end, freshly isolated tissue was fixed for 90 min at 4˚C in PBS containing 1% formaldehyde, 0.2% glutaraldehyde, and 0.02% NP-40. Tissue was washed in ice-cold PBS subsequently and incubated overnight in the dark using PBS containing 5 mM $K_3Fe(CN)_6$, 5 mM $K_4Fe(CN)_6$, 2 mM $MgCl_2$, 1 mg/ml X-Gal, and 0.02% NP-40. After X-Gal staining, tissue was postfixed in 4% buffered formaldehyde in PBS and processed as previously described. Counterstaining of sections was performed with nuclear fast red.

## Cell culture and transfection with *siRNA*

LS180 colorectal cancer cells (ATCC CL187) and HCT116 colorectal cancer cells (ATCC CCL247) were grown in Dulbecco's modified Eagle's medium (DMEM) with 10% fetal calf serum (FCS) and 1% penicillin/streptomycin under standard culture conditions.

Predesigned siRNA's were ordered (siMRPL54, Silencer® select, Thermofisher Hs.356578, # 4427037; siControl, Silencer® select negative controle No1, Thermofisher, #4390843). Cells were transfected after plating in a 24wells plate using Lipofectamine® according to the manufacturer's protocols.

## RNA isolation

For gene expression experiments in cells, *mRNA* isolation was performed 48 hours after transfectionn using the Bioline ISOLATE II RNA Mini kit (BIO-52073, Bioline) according to the manufacturer's instructions.

## cDNA synthesis, and quantitative RT-PCR

Synthesis of cDNA was performed using 1 µg of purified *RNA* using Revertaid reverse transcriptase according to protocol (Fermentas, Vilnius, Lithuania). Quantitative RT-PCR was performed using sensifast SYBR No-ROX Kit (GC-biotech Bio-98020) according to manufacturer's protocol on a BioRad iCycler. v). Relative gene expression was calculated using the LinReg method. For primer sequences see Table 1.

## Enzymatic assay for glucose, L-lactate and pyruvate

Enzymatic assays for glucose and L-lactate were performed as described, using a CLARIOstar microplate reader (BMG LABTECH) [30]. Glucose was measured using a colorimetric assay with glucose oxidase. L-Lactate was measured with lactate dehydrogenase (Roche). Results were normalized to cell count.

## Microarray analysis

For mRNA microarray analysis, a published dataset was used (GSE143509) and analysed using R2 software [31].

## Statistics

Statistical analysis was performed using GraphPad Prism version 9.1.0 (GraphPad Software, San Diego, California USA, www.graphpad.com). All values are depicted as the mean ± standard error of the mean (S.E.M.). In experiments comparing two groups, statistical significance was analysed using Student's *t*-test. Differences were considered statistically significant at $P < 0.05$.

## Results

To analyze the role of MRPL54 in colorectal tumorigenesis, we first analyzed mRNA expression of a panel of mitochondrial ribosomal proteins from mRNA microarrays. These arrays had been generated of murine intestinal epithelial organoids with mutations that are known to drive development of invasive adenocarcinomas from normal intestinal epithelium, a mutational sequence known as the adenoma to carcinoma sequence (*Apc, KrasG12D, Smad4, P53,* Fig 1) [31,32]. Throughout the adenoma to carcinoma sequence, we observed that a large number of these mitochondrial ribosomal proteins were differentially expressed. Focusing on *Mrpl54*, we observed strongly increased expression over the course of the adenoma to carcinoma sequence, most distinctly from the stage of *Apc* and *Kras* mutations. Together these analyses show that during development of colorectal neoplasias, expression of genes involved in the OXPHOS complex are strongly altered and may therefore play a role in this process.

**Table 1. List of primers used for RT-qPCR.**

| Gene | Forward primer | Reverse primer |
|---|---|---|
| *B-ACTIN* | AGAGCTACGAGCTGCCTGAC | AGCACTGTGTTGGCGTACAG |
| *GAPDH* | CCATGTTCGTCATGGGTGTG | GGTGCTAAGCAGTTGGTGGTG |
| *MRPLI54* | TGGGCGTCAACATCTACAAGG | CTCCAGGGTCTTTGGGGGA |
| *LGR5* | AATCCCCTGCCCAGTCTC | CCCTTGGGAATGTATGTCAGA |
| *AXIN2* | TCTGGTGCAAAGACATAGCCA | AGTGTGAGGTCCACGAAAC |
| *ASCL2* | CGCCTACTCGTCGGACGACAG | GCCGCTCGCTCGGCTTCCG |

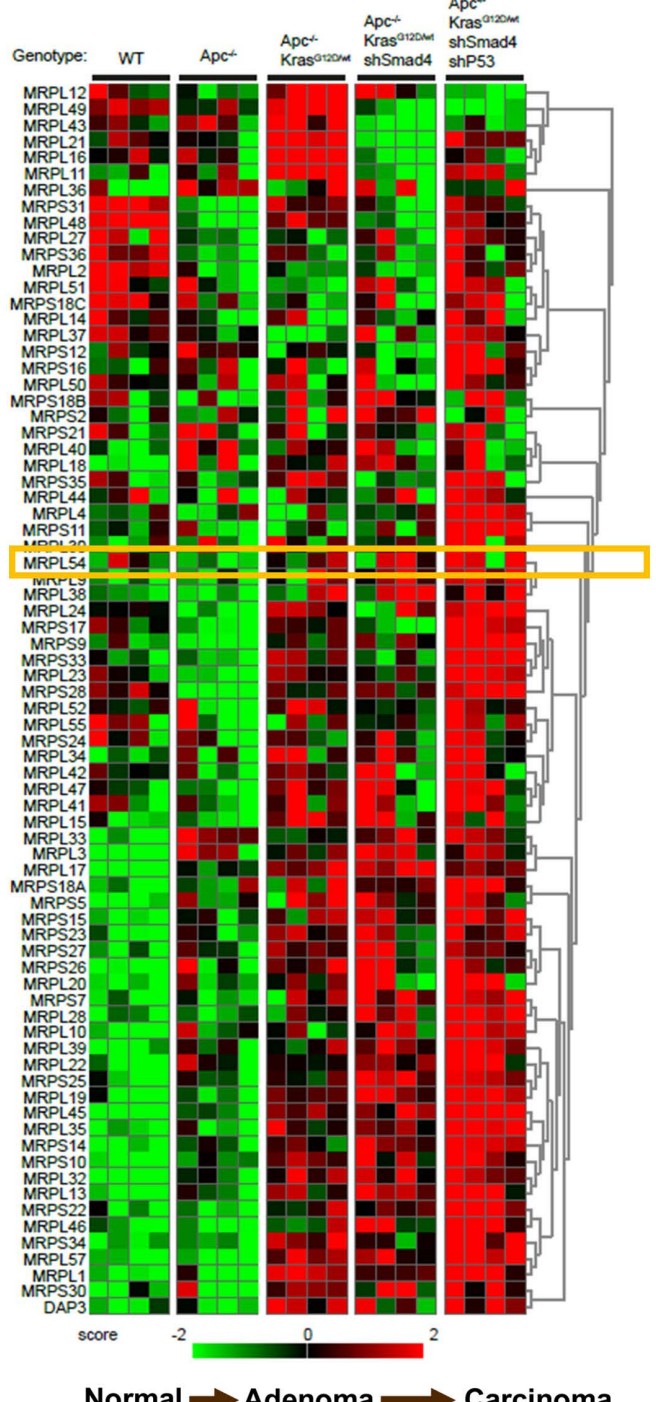

**Fig 1. Micro-array analysis of MRPs.** Heatmap of micro-array mRNA expression containing Mitochondrial Ribosomal Proteins (MRPs) genes in indicated mouse organoids (GSE143509).

We next analyzed whether reducing *Mrpl54* would specifically inhibit adenoma formation. Cells with an *Apc* mutation have increased proliferation and a high energy demand, likely requiring ATP generation via the OXPHOS complex. Genetically reducing a specific mitoribosomal protein, important for the biogenesis of the OXPHOS complex, may therefore challenge the growth of highly proliferative cells To this end we crossed whole body heterozygous *Mrpl54* mice with *VillinCre^ERT2^-Apc^+/fl^* mice. The model of *Mrpl54^+/-^* mice is extensively described recently and exhibits 50% reduced expression of *Mrpl54* transcript throughout different tissues [21].

In *VillinCre^ERT2^-Apc^+/fl^* mice, treatment with tamoxifen results in heterozygous deletion of tumor suppressor gene *Apc* specifically from the small and large intestinal epithelium which in turn results in development of multiple intestinal adenomas several weeks later [23–27].

These *VillinCre^ERT2^-Apc^+/fl^-Mrpl54^+/-^* mice, further referred to as *A-Mrpl54^HET^* were analyzed and compared to *VillinCre^ERT2^-Apc^+/fl^ Mrpl54^+/+^* littermate controls (*A-Mrpl54^WT^*).

Analysis of weight of *A-Mrpl54^HET^* and *A-Mrpl54^WT^* mice did not reveal a difference between these genotypes (S1A Fig). When mice were sacrificed at the age of 6 months, a large number of intestinal adenomas had developed in the small and large intestine of both *A-Mrpl54^HET^* and *A-Mrpl54^WT^*. We confimed equal efficacy of recombination of the conditional *Apc* allele from the intestinal epithelium by visualization of LacZ, that was expressed in all recombined cells (Fig 2A). Macroscopic quantification of the adenoma numbers exhibited no difference between *A-Mrpl54^HET^* and *A-Mrpl54^WT^* animals (Fig 2B). Adenomas in the small intestines were categorized by size, and subdivided in microadenomas (consisting of single crypts), small adenoma (1–3 crypts) and large adenoma (>3 crypts) (Fig 2C). The distribution of microadenomas, small adenomas and large adenomas was comparable between *A-Mrpl54^HET^* and *A-Mrpl54^WT^* (Fig 2D). Microscopic quantification of the small intestine, divided in the proximal, middle and distal parts of the small intestine, showed equal distribution of adenomas between *A-Mrpl54^WT^* and *A-Mrpl54^HET^* (S1B Fig). The colon displayed no difference in adenoma formation between the two genotypes (Fig 2E).

These results demonstrate that heterozygous deletion of *Mrpl54* does not result in a difference in *Apc* mutated adenoma formation and that both adenoma initiation as well as adenoma growth were equal between the two groups.

We next assessed phenotypical differences in proliferation or apoptosis between the two genotypes. No differences were seen in the number of proliferating cells in the crypt of small intestines of *A-Mrpl54^WT^* mice and *A-Mrpl54^HET^* mice (Fig 3A-B). Moreover, cleaved caspase-3, as a marker of apoptosis, was not different between the small intestines of the *A-Mrpl54^HET^* and *A-Mrpl54^WT^* mice *(Fig 3A*,C). In conclusion, in accordance with results on adenoma numbers and growth in *A-Mrpl54^WT^* and *A-Mrpl54^HET^* mice, we found no differences in proliferation or apoptosis.

Previously, in healthy mice under homeostatic conditions, genetic manipulation of *Mrpl54* expression did not result in compelling metabolic changes [21]. We showed that in a hyperproliferative context, using a heterozygous *Apc* mutation, heterozygous *Mrpl54* expression did not influence adenoma formation. To study reduced levels of Mrpl54 on malignantly transformed colorectal cells, we used two human colorectal cancer cell lines, HCT116 and LS180, and tranfected these cells with small interfering RNA (*siRNA*) targeting *MRPL54*. This resulted in cells with ≥85% reduction of *MRPL54 mRNA* (S2A Fig). In concordance with the adenoma study, no difference in stem cell marker expression was found between colorectal cancer cells with *siControl* or *siMRPL54* (S2B Fig). Subsequently, we evaluated glucose consumption and lactate production in cells exhibiting either normal or diminished *MRPL54* expression. Downregulation of *MRPL54 mRNA* in colorectal cancer cells did not affect these basic metabolic assays.

## Discussion

Previous work has shown that the mitochondrial ribosomal protein 54 (MRPL54) is critical for mitochondrial energy metabolism, though *Mrpl54 heterozygous* knockout mice have no difference in body composition or metabolism compared to wild type mice during homeostasis [20,21]. Because cancer cells require higher levels of energy, which is a mitochondrial tenet, we hypothesized that *Mrpl54* heterozygosity would hamper proliferation of intestinal adenoma cells. We find

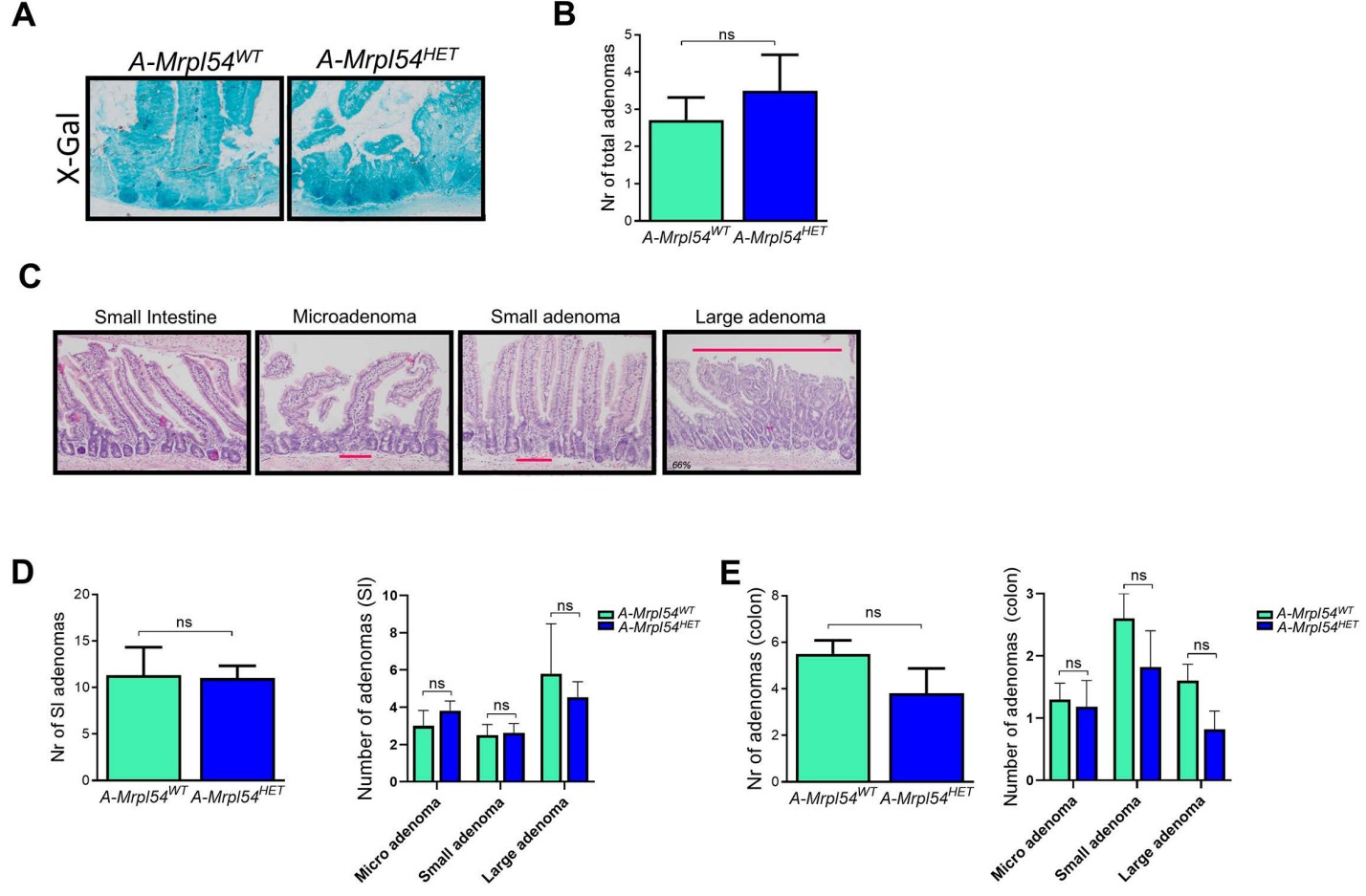

**Fig 2. Mrpl54 heterozygosity does not alter adenoma formation in ApcHET mice. (A)** LacZ staining on intestines of mice showing recombination after induction of Cre-mediated recombination with tamoxifen. **(B)** Number of intestinal adenomas, counted macroscopically (A-Mrpl54WT N = 7, A-Mrpl54HET N = 8). **(C)** Representative pictures of H&E staining of normal small intestine, microadenoma (single crypt), small adenoma (1-3 crypts), large adenoma (>3 crypts). **(D)** Quantification of small intestinal adenomas, counted microscopically after H&E staining (A-Mrpl54WT N = 10, A-Mrpl54HET N = 11). **(E)** Quantification of colon adenomas, counted microscopically after H&E staining (A-Mrpl54WT N = 10, A-Mrpl54HET N = 11). ns = non-significant SI = small intestine.

no evidence for the hypothesis that reduced expression of *Mrpl54*, either through genetic heterozygosity in mice or by the use of *siRNA* in colorectal cancer cells, influences adenoma formation or induces significant metabolic alterations in hyperproliferative models.

Our experiments demonstrate that *Apc-Mrpl54 (A-Mrpl54)* heterozygous intestines exhibit no significant differences in adenoma initiation or growth. Moreover, aspects of cellular functioning such as proliferation or apoptosis are unaltered in *A-Mrpl54HET* cells compared to *A-Mrpl54WT* cells. The used *Apc* mouse model harbors a heterozygous mutation in tumor suppressor gene, though the murine organoid micro-array data showed expression of *Mrpl54* in the context of multiple intestinal carcinoma mutations [31]. However, analysis of micro-array data sets of human colorectal cancer that are publically available, showed no relation between *MRPL54* expression and colorectal cancer. These results are consistent with findings in the human protein atlas that show no correlation between MRPL54 expression and prognosis in colorectal cancer. These findings correlate with our findings that *Mrpl54 heterozygosity* does not influence murine adenomagenesis. We observe an important discrepancy between the observational micro-array data that show correlation between progression

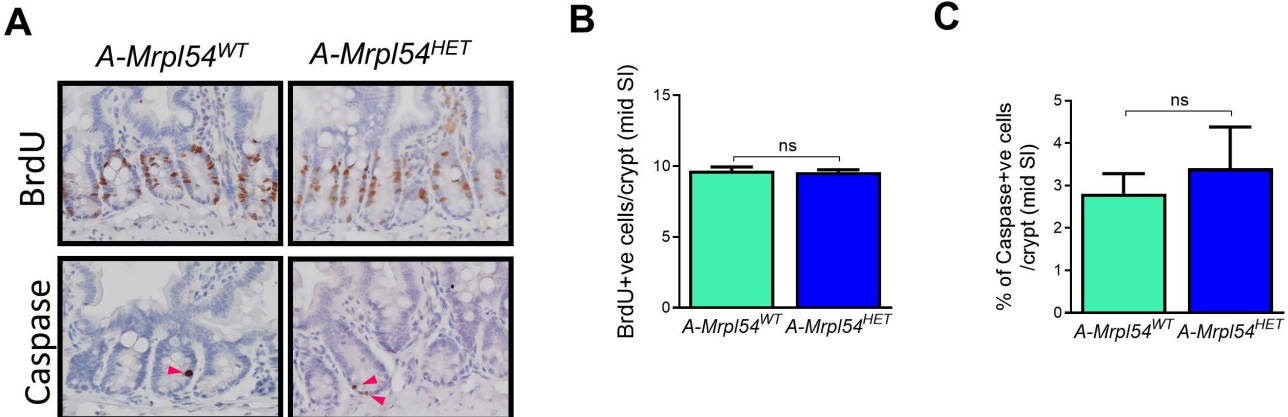

**Fig 3. Microscopic analysis of proliferation and apoptosis. (A)** Representative BrdU and caspase staining of indicated genotypes. **(B)** Quantification of mean BrdU+ve cells per crypt (A-Mrpl54WT N = 10, A-Mrpl54HET N = 11). **(C)** Quantification of % Caspase+ve cells per crypt (A-Mrpl54WT N = 10, A-Mrpl54HET N = 11). ns = non-significant.

of tumor mutations and expression of *Mrpl54* on the one hand whereas reduction of *Mrpl54* did not yield reduced tumorigenesis in the murine model. Potentially this stems from compensation of *Mrpl54* levels upon heterozygous reduction or due to redundancy with other *Mrpls*. In addition, redundancy may have explained lack of phenotypical differences in colorectal cancer lines.

The *Mrpl54* heterozygous knockout mice is extensively described by Reid et al. and showed a bodywide reduction of *Mrpl54* gene expression by 50%. Unfortunately, experiments to confirm decreased protein levels of MRPL54 were unsuccessful by the combined efforts of both research groups using both western blotting and mass spectrometry. The observation that pups homozygous for the *Mrpl54* mutation did not survive, is in clear agreement with a dysfunctional *Mrpl54* allele.

Overall, our results indicate that reduced *Mrpl54 mRNA* expression does not lead to changes in intestinal adenoma formation or global cellular metabolism.

## Supporting information

**S1 Fig. Mrpl54 and Apc heterozygosity do not affect incremental body weight.** (A) Weight curves of indicated genotypes (A-Mrpl54WT N = 10, A-Mrpl54HET N = 11). (B) Quantification of small intestinal adenomas, counted microscopically after H&E staining, subdived by localisation (A-Mrpl54WT N = 10, A-Mrpl54HET N = 11). ns = non-significant.
(TIF)

**S2 Fig. Diminished Mrpl54 expression does not alter stemness or metabolism in colorectal cancer cells.**
(TIF)

**S1 Data. Rawdata.**
(XLSX)

## Author contributions

**Conceptualization:** Claudia N. Spaan, Eileen Daniels, Wouter L. Smit, Vanesa Muncan, Riekelt H. Houtkooper, Jarom Heijmans.

**Data curation:** Claudia N. Spaan, Eileen Daniels, Wouter L. Smit, Joana Silva.

**Formal analysis:** Claudia N. Spaan, Eileen Daniels, Wouter L. Smit, Vanesa Muncan, Riekelt H. Houtkooper, Jarom Heijmans, Ruben J. de Boer, S. Meisner, Jacqueline L.M. Vermeulen.

**Funding acquisition:** Riekelt H. Houtkooper, Jarom Heijmans.

**Investigation:** Claudia N. Spaan, Wouter L. Smit, Ruben J. de Boer, S. Meisner, Jacqueline L.M. Vermeulen.

**Methodology:** Claudia N. Spaan, Eileen Daniels, Joana Silva.

**Project administration:** Vanesa Muncan, Riekelt H. Houtkooper, Jarom Heijmans.

**Resources:** Joana Silva.

**Supervision:** Vanesa Muncan, Riekelt H. Houtkooper, Jarom Heijmans.

**Validation:** Claudia N. Spaan, Eileen Daniels, Vanesa Muncan.

**Visualization:** Claudia N. Spaan, Eileen Daniels.

**Writing – original draft:** Claudia N. Spaan, Eileen Daniels, Riekelt H. Houtkooper, Jarom Heijmans.

**Writing – review & editing:** Claudia N. Spaan, Eileen Daniels, Joana Silva, Vanesa Muncan, Riekelt H. Houtkooper, Jarom Heijmans.

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
