## [Decision Letter · Decision Letter 0]

7 May 2025

Dear Dr. Heijmans,

Thank you for submitting your manuscript to PLOS ONE. After careful consideration, we feel that it has merit but does not fully meet PLOS ONE’s publication criteria as it currently stands. Therefore, we invite you to submit a revised version of the manuscript that addresses the points raised during the review process.

The manuscript set to investigate the plausible role of the mitochondrial ribosomal protein MRPL54 in colorectal tumorigenesis. Using micro-arrays of intestinal organoids with multiple oncogenic mutations, the authors identified upregulation of MRPL54 and other MRPs proteins and used previously established (by their group) *Mrpl54* heterozygous mouse model of intestinal carcinoma (Apc-mutant). No detectable effect of MRPL54 heterozygosity was observed compared to the parental mouse model of intestinal carcinoma.

The study is based solely on in vivo data and lacks mechanical insight. A few experiments need to be performed to make the story complete.

Reduced MRPL54 protein expression or function in the model has to be confirmed.

A role of MPRL54 protein in tumorigenesis had been established using limited datasets (microarray and gross adenoma counts). No analysis of patient-derived CRC data for MRPL54 expression or prognostic relevance had been accomplished.

A few in vitro functional assays in CRC cell lines (e.g., siRNA knockdown, metabolic flux analysis, ROS measurement) would help to confirm the rationale for the study.

Furthermore, functional redundancy among multiple MRPs is not addressed. This is particularly important for metabolically flexible colorectal cancers that use both glycolysis and OXPHOS as energy sources.

Minor comments:

The manuscript lacks clarity in both writing and data interpretation. For example, the abstract does not clearly reflect the study rationale.

Discussion needs to be edited.

We look forward to receiving your revised manuscript.

Kind regards,

Irina V. Lebedeva, Ph.D.

Academic Editor

PLOS ONE

Additional Editor Comments (if provided):

Reviewers' comments:

Reviewer's Responses to Questions

**Comments to the Author**

1. Is the manuscript technically sound, and do the data support the conclusions?

Reviewer #1: Yes

Reviewer #2: Yes

Reviewer #3: Yes

2. Has the statistical analysis been performed appropriately and rigorously?

Reviewer #1: Yes

Reviewer #2: Yes

Reviewer #3: Yes

3. Have the authors made all data underlying the findings in their manuscript fully available?

Reviewer #1: Yes

Reviewer #2: Yes

Reviewer #3: Yes

4. Is the manuscript presented in an intelligible fashion and written in standard English?

Reviewer #1: Yes

Reviewer #2: Yes

Reviewer #3: Yes

Reviewer #1: Apart from few gramatical errors, the manuscript is scientifically well written. The methods section and results section is well detailed appropriate for the conducted study. The first paragraph of discussion reads more like results section and need to be edited.

Reviewer #2: The authors investigate the potential role of the mitochondrial ribosomal protein MRPL54 in colorectal tumorigenesis. Citing increased MRPL54 expression in colorectal cancer and the known function of mitochondrial ribosomal proteins (MRPs) in metabolic regulation, they hypothesize that reduced expression of Mrpl54 (via heterozygosity) would suppress adenoma formation in a genetically engineered mouse model with Apc mutations. However, they report that Mrpl54 heterozygosity has no detectable effect on adenoma formation or progression, and thus conclude that MRPL54 is not essential for Apc-driven colorectal tumor development.

Major Comments

1. Lack of Mechanistic Depth and Novel Insight

While MRPL54 has been identified in transcriptomic datasets as upregulated in various cancers, the manuscript does not offer new mechanistic insight into its role or lack thereof in colorectal cancer. The negative finding (no phenotype in heterozygous mice) is not inherently unpublishable, but it must be contextualized with mechanistic rigor, which is absent here.

2. Insufficient Justification of Experimental Strategy

The central hypothesis—that MRPL54 is essential for adenoma formation is not adequately supported or tested:

Heterozygosity may not be sufficient to significantly impair mitochondrial translation.

There is no confirmation of reduced MRPL54 protein expression or function in the model.

Functional redundancy among MRPs (of which there are ~82) is not addressed.

3. Overly Simplistic Assumption Regarding Metabolism and Tumorigenesis

The authors propose that reducing a single MRP should alter tumorigenesis, but colorectal cancers are metabolically flexible, using both glycolysis and oxidative phosphorylation (OXPHOS). It is therefore not surprising that reducing one MRP does not disrupt tumor initiation. This limitation weakens the conceptual novelty of the work.

4. Incomplete Exploration of MRPL54’s Role

The study relies heavily on:

A single in vivo model (Apc-mutant mice)

Limited datasets (microarray and gross adenoma counts)

No in vitro functional assays (e.g., siRNA knockdown, metabolic flux analysis, ROS measurement)

No analysis of patient-derived CRC data for MRPL54 expression or prognostic relevance.

As such, the manuscript does not sufficiently explore whether MRPL54 has any functional role in tumor metabolism, survival, or progression.

5. Presentation Issues

The manuscript lacks clarity in both writing and data interpretation. Several sections are grammatically inconsistent or incomplete.

To strengthen future studies, the authors could:

Confirm MRPL54 knockdown at the protein and functional level in their model.

Perform functional assays in CRC cell lines (e.g., siRNA/CRISPR knockdown with proliferation, OXPHOS, and ROS readouts).

Analyze clinical datasets (e.g., TCGA) to correlate MRPL54 expression with patient survival and tumor stage.

Explore compensatory responses by other MRPs or mitochondrial stress response pathways.

Investigate metabolic consequences of MRPL54 loss using Seahorse or mass spectrometry–based metabolomics.

Reviewer #3: This study investigated whether reduced levels of mitochondrial ribosomal protein MRPL54 affect the tumorigenic characteristics of colorectal cancer in mice. MRPL54 was chosen as a target protein because its expression levels correlate with the progression and proliferation stages of this type of cancer, as shown by micro-array mRNA expression results. The authors aimed to test whether reducing MRPL54 levels would inhibit adenoma formation due to its importance in the biogenesis of OXPHOS proteins. Results from the study reveal that deletion of MRPL54 does not alter intestinal adenoma formation in mice. Both WT and KO animals had similar body weight and characteristics at 6 months of age, before euthanasia. Similar numbers of adenomas were recorded in both groups of animals.

Overall, this is a concise and clear study. The conclusions are well supported by the results, and data presentations are clear. However, several typos throughout the manuscript need to be corrected before publication.

**Do you want your identity to be public for this peer review?** For information about this choice, including consent withdrawal, please see our Privacy Policy

Reviewer #1: **Yes:** Silpa Gampala

Reviewer #2: No

Reviewer #3: No

---

## [Author Response · Author response to Decision Letter 1]

7 Jan 2026

1. The study is based solely on in vivo data and lacks mechanical insight. A few experiments need to be performed to make the story complete.

Reduced MRPL54 protein expression or function in the model has to be confirmed.

This research continued the work of Reid and Daniels, using the Mrpl54 bodywide heterozygous mouse model described in Reid et al, 2023. In that study they showed 50% reduction of Mrpl54 gene expression in multiple organs in MRPL54 heterozygous (genotyped) mice. The observation that pups homozygous for the Mrpl54 mutation did not survive as reflected by the Mendelian ratio, is in agreement with the dysfunctional Mrpl54 allele.

Tissue of heterozygous and wildtype Mrpl54 mice was used to perform western blots to measure protein expression. Unfortunately, despite using multiple antibodies, these were technically unsuccessful because of nonspecific binding. Our co-authors also tried mass spectrometry, but were not able to obtain a high enough signal.

In our manuscript, we included explanation of these results and clarified the missing data of protein expression.

2. A role of MPRL54 protein in tumorigenesis had been established using limited datasets (microarray and gross adenoma counts). No analysis of patient-derived CRC data for MRPL54 expression or prognostic relevance had been accomplished.

We agree with the reviewer’s suggestion to investigate multiple publicly available human colorectal cancer (CRC) datasets. We have analyzed these datasets for Mrpl54 expression and found no correlation within the human CRC data. Additionally, we observed no association between MRPL54 and colorectal cancer survival in the Human Protein Atlas. These differences between the organoid and human datasets can be attributed to the inherent complexity of the models used. However, we believe these findings are consistent with the results from our murine experiment. This point has been addressed in our discussion. Due to limitations in word count and figures we chose not to add results of CRC dataset analysis to the result section, but this could be added if the reviewer deems this preferrable.

3. A few in vitro functional assays in CRC cell lines (e.g., siRNA knockdown, metabolic flux analysis, ROS measurement) would help to confirm the rationale for the study.

We have added a supplementary figure (2), including experiments with two different human colorectal cancer cell lines transfected with highly specific small interfering RNA targeting MRPL54 as well as a control. In these cells, we demonstrate that although MRPL54 RNA expression is reduced, this does not impact the expression of stem cell markers nor affect metabolic assays as glucose consumption and lactate production.

4. Furthermore, functional redundancy among multiple MRPs is not addressed. This is particularly important for metabolically flexible colorectal cancers that use both glycolysis and OXPHOS as energy sources.

We agree with the reviewer and have now addressed this in our discussion.

5. Minor comments:

The manuscript lacks clarity in both writing and data interpretation. For example, the abstract does not clearly reflect the study rationale.

Discussion needs to be edited.

To address this comment we have made textual changes and we have asked colleagues that are native English speakers to give comments on our writing. Textual changes have been made throughout the manuscript.

We have addressed these style requirements.

We have addressed this.

Reviewer 1

1. The first paragraph of discussion reads more like results section and need to be edited.

We have revised the discussion and incorporated feedback provided by the editor and reviewers. We feel the revised version of the manuscript has a more comprehensive discussion part compared to the original manuscript.

Reviewer 2

1. Lack of Mechanistic Depth and Novel Insight.

While MRPL54 has been identified in transcriptomic datasets as upregulated in various cancers, the manuscript does not offer new mechanistic insight into its role or lack thereof in colorectal cancer. The negative finding (no phenotype in heterozygous mice) is not inherently unpublishable, but it must be contextualized with mechanistic rigor, which is absent here.

We have expanded our discussion and include finding on Mrpl54 expression in human datasets, as well as new experiments involving a more profound knockdown of Mrpl54 in colorectal cancer cells. We have addressed the fenomenon of redundancy in the discussion.

2. Insufficient Justification of Experimental Strategy

The central hypothesis—that MRPL54 is essential for adenoma formation is not adequately supported or tested:

Heterozygosity may not be sufficient to significantly impair mitochondrial translation.

There is no confirmation of reduced MRPL54 protein expression or function in the model.

Functional redundancy among MRPs (of which there are ~82) is not addressed.

We agree with the reviewers comments, which have also been addressed by the editor, and we have include redundancy in our revised manuscript. As previousily mentioned, we have added experiments with siRNA’s in cell lines to achieve a more robust knockdown have a more profound knockdown. Furthermore, we emphasize in the manuscript previous experiments with the same Mrpl54 mouse model (Reid et al.), its findings and earlier efforts to confirm decreased MRPL54 protein expression.

3. Overly Simplistic Assumption Regarding Metabolism and Tumorigenesis

The authors propose that reducing a single MRP should alter tumorigenesis, but colorectal cancers are metabolically flexible, using both glycolysis and oxidative phosphorylation (OXPHOS). It is therefore not surprising that reducing one MRP does not disrupt tumor initiation. This limitation weakens the conceptual novelty of the work.

4. Incomplete Exploration of MRPL54’s Role

The study relies heavily on:

A single in vivo model (Apc-mutant mice)

Limited datasets (microarray and gross adenoma counts)

No in vitro functional assays (e.g., siRNA knockdown, metabolic flux analysis, ROS measurement)

No analysis of patient-derived CRC data for MRPL54 expression or prognostic relevance.

As such, the manuscript does not sufficiently explore whether MRPL54 has any functional role in tumor metabolism, survival, or progression.

We have addressed these comments in our manuscript and have written a more detailed explanation of our revisions in the previous comments of the editor and reviewers.

5. Presentation Issues

The manuscript lacks clarity in both writing and data interpretation. Several sections are grammatically inconsistent or incomplete.

To address this comment we have made textual changes and we have asked colleagues that are native English speakers to give comments on our writing. Textual changes have been made throughout the manuscript.

Reviewer 3

1. Overall, this is a concise and clear study. The conclusions are well supported by the results, and data presentations are clear. However, several typos throughout the manuscript need to be corrected before publication.

We have adressed this issue, and textual changes have been made throughout the manuscript.

---

## [Decision Letter · Decision Letter 1]

18 Feb 2026

Reduced levels of mitochondrial ribosomal protein MRPL54 does not alter Apc related adenoma formation

PONE-D-25-10633R1

Dear Dr. Heijmans,

We’re pleased to inform you that your manuscript has been judged scientifically suitable for publication and will be formally accepted for publication once it meets all outstanding technical requirements.

Kind regards,

Irina V. Lebedeva, Ph.D.

Academic Editor

PLOS One

Additional Editor Comments (optional):

Reviewers' comments:

Reviewer's Responses to Questions

**Comments to the Author**

Reviewer #1: All comments have been addressed

Reviewer #2: All comments have been addressed

Reviewer #3: All comments have been addressed

2. Is the manuscript technically sound, and do the data support the conclusions?

Reviewer #1: Yes

Reviewer #2: Yes

Reviewer #3: Yes

3. Has the statistical analysis been performed appropriately and rigorously?

Reviewer #1: Yes

Reviewer #2: Yes

Reviewer #3: Yes

4. Have the authors made all data underlying the findings in their manuscript fully available?

Reviewer #1: Yes

Reviewer #2: Yes

Reviewer #3: Yes

5. Is the manuscript presented in an intelligible fashion and written in standard English?

Reviewer #1: Yes

Reviewer #2: Yes

Reviewer #3: Yes

Reviewer #1: The authors have adequately addressed the reviewer comments, and I recommend acceptance of the revised manuscript.

Reviewer #2: all queries answered, authors have corrections. Abstract seems reduced but reads all right. Figures can be refined for the publication.

Reviewer #3: (No Response)

**Do you want your identity to be public for this peer review?** For information about this choice, including consent withdrawal, please see our Privacy Policy

Reviewer #1: **Yes:** Silpa Gampala

Reviewer #2: **Yes:** Narendra Sankpal

Reviewer #3: No

---

## [Editor Report · Acceptance letter]

PONE-D-25-10633R1

PLOS One

Dear Dr. Heijmans,

I'm pleased to inform you that your manuscript has been deemed suitable for publication in PLOS One. Congratulations! Your manuscript is now being handed over to our production team.

Kind regards,

on behalf of

Dr. Irina V. Lebedeva

Academic Editor

PLOS One